# Impact of Novel Agents on Allogeneic Hematopoietic Cell Transplantation in Patients with T-Cell Lymphomas

**DOI:** 10.3390/cells14171306

**Published:** 2025-08-23

**Authors:** Yoshitaka Inoue, Jun-ichirou Yasunaga

**Affiliations:** 1Blood and Marrow Transplant Program, Division of Hematology and Oncology, Penn State Cancer Institute, Hershey, PA 17033, USA; 2Cancer Institute, Penn State College of Medicine, Hershey, PA 17033, USA; 3Department of Hematology, Kumamoto University, Kumamoto 860-8556, Japan; jyasunag@kumamoto-u.ac.jp

**Keywords:** T-cell lymphomas, allogeneic hematopoietic cell transplantation, novel agents, mogamulizumab, brentuximab vedotin, lenalidomide, histone deacetylase inhibitors (HDACis), enhancer of zeste homolog 1/2 (EZH1/2) inhibitors

## Abstract

T-cell lymphomas (TCLs) are generally associated with a poorer prognosis compared to B-cell lymphomas, and allogeneic hematopoietic cell transplantation (allo-HCT) is often considered for eligible patients. One of the primary reasons for the inferior outcomes in TCLs has been the lack of effective novel agents for many years, resulting in a continued reliance on traditional cytotoxic chemotherapy regimens. However, over the past decade, several novel agents with promising efficacy against TCLs have been developed. Notably, many of these agents not only exert direct anti-tumor effects but also modulate host immune function, raising clinical questions regarding the optimal integration of these agents with allo-HCT. In this review, we aim to summarize how the use of novel agents that are approved for the treatment of TCLs—such as mogamulizumab, brentuximab vedotin, lenalidomide, histone deacetylase inhibitors, enhancer of zeste homolog inhibitors, and immune checkpoint inhibitors—before or after allo-HCT may impact transplantation outcomes in patients with TCLs.

## 1. Introduction

T-cell lymphomas (TCLs) are a heterogeneous group of non-Hodgkin lymphomas (NHLs) that originate from mature T cells or natural killer (NK) cells, and include both systemic/peripheral T-cell and NK/T-cell lymphomas (PTCL/NKTCL) and primary cutaneous T-cell lymphomas (CTCL). They are relatively rare, accounting for approximately 10–15% of all NHLs in Western countries, but are more prevalent in parts of Asia, where the frequency may reach up to 20% depending on geographic and ethnic factors [1,2,3]. Among the various subtypes, the most common include peripheral T-cell lymphoma, not otherwise specified (PTCL-NOS), angioimmunoblastic T-cell lymphoma (AITL), anaplastic large cell lymphoma (ALCL), and adult T-cell leukemia/lymphoma (ATL), as well as mycosis fungoides (MF) and Sézary syndrome (SS) among cutaneous types, each with distinct clinical and pathological features [4,5].

Compared to B-cell lymphomas, TCLs are generally associated with significantly poorer outcomes. For instance, while diffuse large B-cell lymphoma (DLBCL) treated with standard immunochemotherapy (e.g., R-CHOP) can achieve 5-year overall survival (OS) of approximately 60–70%, most TCL subtypes treated with CHOP-based regimens have reported 5-year OS rates below 35% [6]. Median progression-free survival (PFS) in relapsed or refractory PTCL remains less than 6 months in many series [7,8]. One of the major contributing factors to these poor outcomes has been the historical lack of effective targeted therapies for TCLs, leading to continued reliance on cytotoxic chemotherapy with limited efficacy.

Unlike autologous transplantation, allogeneic hematopoietic cell transplantation (allo-HCT) offers a graft-versus-lymphoma (GVL) effect, which may provide sustained disease control even in high-risk patients [9]. Allo-HCT has emerged as a potentially curative strategy for patients with TCL, particularly those with relapsed or refractory disease [10,11]. For example, in advanced-stage MF and SS, multiple prospective and retrospective studies have demonstrated the efficacy of allo-HCT, with 3-year overall survival rates reaching 40–60% [12,13,14]. In AITL, an analysis of 249 patients with refractory disease or relapse after autologous transplantation reported 4-year PFS and OS rates of 49% and 56%, respectively [15]. Moreover, in patients with ATL, up-front allo-HCT is strongly recommended as standard therapy [16,17]. However, the outcomes of allo-HCT in patients with TCL remain suboptimal overall, with 3-year OS reported to range from 30–60% in most retrospective series, and non-relapse mortality (NRM) ranging from 20–30%, particularly in heavily pretreated populations or those undergoing transplantation with active disease [18].

Over the past decade, there has been growing interest in the use of novel agents for the treatment of TCLs. These include monoclonal antibodies (e.g., mogamulizumab; MOG [19], brentuximab vedotin; BV [20]), immune modulators (e.g., lenalidomide [21]), histone deacetylase (HDAC) inhibitors (e.g., vorinostat [22], romidepsin [23], belinostat [24], chidamide (also known as tucidinostat) [25]), enhancer of zeste homolog 1/2 (EZH1/2) inhibitors (e.g., valemetostat [26]), and immune checkpoint inhibitors (ICIs) (e.g., pembrolizumab [27]). In the context of transplantation for TCLs, achieving optimal disease control prior to allo-HCT is one of the most critical factors of outcome [15,28]. Given the activity of these novel agents in TCLs, they are increasingly being utilized as salvage therapy for chemotherapy-refractory disease or as a bridge to allo-HCT. Furthermore, unlike conventional cytotoxic agents, some of these drugs have relatively limited myelosuppression, making them feasible for use as post-transplant maintenance therapy aimed at reducing the risk of relapse. Notably, several of these agents not only demonstrate direct anti-tumor activity but also influence host immune responses, raising important questions about their optimal timing and sequencing with allo-HCT.

In this review, we aim to summarize the current evidence regarding the use of these novel agents before or after allo-HCT in patients with TCLs, and how their integration may influence transplantation outcomes.

## 2. Antibody Drugs

### 2.1. Mogamulizumab (MOG)

MOG, an anti-C-C chemokine receptor type 4 (CCR4) antibody, was developed in Japan as a therapeutic agent for ATL, which frequently and highly expresses CCR4 [29]. CCR4 is a type of chemokine receptor primarily expressed on regulatory T cells (Tregs) and T-helper type 2 (Th2) cells under normal physiological conditions [30,31]. It binds to the chemokines C-C motif chemokine ligand 17 and C-C motif chemokine ligand 22, playing a role in T-cell homing to specific tissues [32]. MOG is characterized by a modified Fc region that enhances antibody-dependent cellular cytotoxicity (ADCC) [33]. By binding to CCR4-expressing tumor cells, it induces ADCC and eliminates tumor cells, while also depleting immunosuppressive Tregs, thereby potentially enhancing anti-tumor immunity [34]. In the United States, it was approved by the Food and Drug Administration (FDA) in 2018 following a phase III clinical trial targeting CTCL, specifically MF and SS [35].

#### 2.1.1. Impact of Pre-HCT Use

In allo-HCT for ATL, it was reported that achieving remission at the time of allo-HCT was the most important factor influencing transplant outcomes [36]. The combination of MOG with conventional chemotherapy was shown to improve remission rates [37], and it was initially expected that using MOG prior to allo-HCT would allow patients to undergo allo-HCT in a better disease status. On the other hand, as mentioned earlier, MOG not only targeted tumor cells but also depleted Tregs, which are key players in maintaining immune tolerance after allo-HCT. Therefore, there were concerns that the use of MOG before allo-HCT could lead to Treg depletion and a subsequent increase in the risk of graft-versus-host disease (GVHD).

In response to this controversy, a single-center study reported that patients who received MOG prior to allo-HCT (*n* = 11) had significantly higher NRM (58.2% vs. 12.4%, *p* = 0.001) and poorer OS (31.8% vs. 77.1%, *p* = 0.004) compared to those who did not (*n* = 80) (Figure 1) [38]. A subsequent nationwide study in Japan comparing 82 patients who received MOG prior to allo-HCT with 914 patients who did not similarly demonstrated a significantly increased risk of steroid-refractory GVHD in the MOG group. This was accompanied by a significant increase in NRM (HR 1.93, *p* < 0.01) and overall mortality (HR 1.67, *p* < 0.01) [39]. This study also examined the interval between the last administration of MOG and allo-HCT. Interestingly, it demonstrated that the impact on NRM was attenuated in patients who had an interval of 50 days or more (HR 1.30, *p* = 0.25) compared to those with an interval of less than 50 days (HR 2.02, *p* < 0.01). Based on these reports, it is currently recommended to avoid the use of MOG prior to allo-HCT whenever possible in patients who are planned to undergo allo-HCT [16].

MOG has been reported to be more effective not only in relapsed or refractory cases, but also when used for first-line therapy [40]. The disease can progress rapidly in ATL, and there are often cases in which disease control is difficult to achieve with conventional chemotherapy alone. Then, should transplantation be avoided in patients who have received MOG prior to allo-HCT? Based on the findings from the Japanese nationwide study, although the impact on NRM appears to be attenuated when the interval between the last administration of MOG and allo-HCT is 50 days or more, it is important to note that the risk still remains higher compared to patients who did not receive MOG [39]. Unfortunately, there is no clear consensus on how long the interval should be in order to minimize the adverse impact of MOG on transplant outcomes [41]. We previously reported cases in which allo-HCT was safely performed by ensuring a 2–3 month interval between the last administration of MOG and allo-HCT, and by confirming that the serum concentration of MOG had decreased to certain levels prior to transplantation [42]. Although measurement of the serum MOG level is not commercially available, it may serve as a potential indicator to help assess transplant safety. In patients for whom transplantation is planned, a strategy of using MOG earlier in the treatment course and limiting it to a small number of doses may also be effective (Figure 2) [43]. Furthermore, other case reports have shown that using umbilical cord blood, which is associated with a relatively low risk of GVHD as a donor source, or the use of anti-thymocyte globulin, enabled safe transplantation even after MOG administration [44,45,46,47]. On the other hand, caution is warranted when using post-transplant cyclophosphamide (PTCy), which is now widely used for GVHD prophylaxis. In murine models, the efficacy of PTCy was found to be diminished in settings where Tregs had been depleted [48]. It is also worth noting that in a prospective trial of PTCy for ATL conducted in Japan, patients who had received MOG prior to transplantation were excluded as part of the eligibility criteria [49]. These findings suggest that, by optimizing the donor source and GVHD prophylaxis strategy, it may be possible to safely perform allo-HCT even in patients who have received MOG prior to transplantation.

#### 2.1.2. Impact of Post-HCT Use

Relapse of ATL after allo-HCT is associated with a dismal prognosis, and no standard treatment has been established [50]. However, strategies aimed at enhancing the GVL effect, such as tapering or discontinuing immunosuppressive agents and donor lymphocyte infusion (DLI), have shown potential efficacy in some cases [36,51]. The administration of MOG after allo-HCT is expected to exert not only a direct anti-tumor effect but also an enhanced GVL effect through the suppression of Tregs. Several reports suggest that post-transplant administration of MOG might be effective and relatively safe in certain cases [52,53,54,55].

Compared to administration prior to allo-HCT, why is post-transplant administration of MOG relatively safe? One possible reason is that the interval between allo-HCT and the initiation of MOG therapy was longer than three months in all reported cases, following the period during which immune reconstitution is typically established. Another possible reason may be related to the fact that MOG primarily targets CCR4 highly expressing effector Tregs [56]. We reported that following MOG administration after allo-HCT, the proportion of CD3^+^, CD4^+^, CD25^+^, CD127^dim^, CCR4^high^ cells—representing effector-type Tregs—was markedly reduced, whereas the CCR4^low^ population, corresponding to naïve-type Tregs, was preserved [52]. When MOG is administered after a certain period post-transplant, during which immune reconstitution has progressed, it may suppress effector-type Tregs, while preserving naïve-type Tregs, which are thought to be important for the suppression of GVHD. This may reduce the risk of GVHD. However, reports on the use of MOG in pre- and post-transplant settings are limited to small retrospective series, and further investigations from both basic and clinical perspectives are required.

### 2.2. Brentuximab Vedotin (BV)

BV is an antibody–drug conjugate consisting of an anti-CD30 monoclonal antibody linked to the cytotoxic agent monomethyl auristatin E (MMAE) [57]. Upon binding to CD30-positive cells, BV is internalized, and MMAE is released intracellularly, leading to microtubule disruption and subsequent induction of cell death. In normal human cells, CD30 expression is restricted to activated B cells, activated T cells, and eosinophils, all of which are present in very small proportions [58]. Therefore, CD30 can be considered an appropriate tumor-specific target. Initially developed as a therapeutic agent for Hodgkin lymphoma, which expresses CD30, its indications have since been expanded to include CD30-positive T-cell lymphomas. Clinically significant adverse effects include peripheral neuropathy, which is considered to be caused by MMAE; in addition, myelosuppression and pulmonary toxicity may occur, particularly in combination with other chemotherapeutic agents [59].

#### 2.2.1. Impact of Pre-HCT Use

The use of BV as a bridging therapy to allo-HCT has been reported in numerous studies, particularly in patients with Hodgkin lymphoma [60,61,62]. None of these reports have identified any direct significant negative impact of pre-transplant BV administration on allo-HCT outcomes. Compared to Hodgkin lymphoma, studies evaluating the impact of pre-transplant BV administration as a bridging therapy in patients with TCLs are limited and involve small sample sizes [62,63,64,65,66]. The largest study to date evaluating the impact of pre-transplant BV administration in patients with TCLs was reported by Garciaz et al., involving 26 patients (15 in ALCL, 5 in CTCL, 4 in PTCL-NOS, 1 in enteropathy-associated T-cell lymphoma, and 1 in ATL) [62]. The best response to BV was achieved after 4 cycles, with 17 (65%) patients achieving complete response (CR) and 9 (35%) patients achieving partial response. The median duration from the end of BV therapy to allo-HCT was 44 (range: 7–206) days. In patients with ALCL, they compared outcomes with historical controls who had not received BV, and a significantly higher pre-transplant CR rate was observed in the BV-treated group. Although no significant differences in post-transplant PFS or OS were observed based on the use of BV prior to allo-HCT, this may be attributable to the small sample size. The study found that BV had no adverse effects on engraftment or the incidence of GVHD. Across these reported studies, pre-transplant administration of BV has been associated with favorable treatment responses without a significant increase in post-transplant toxicities, supporting its role as an appropriate bridging strategy prior to allo-HCT.

#### 2.2.2. Impact of Post-HCT Use

To date, there are no comprehensive studies evaluating the use of BV for relapsed TCLs following allo-HCT. Therefore, its efficacy in this setting remains unclear. However, based on data from patients with Hodgkin lymphoma, post-transplant use of BV appears to be relatively safe [67,68,69]. According to these reports, the adverse events associated with post-transplant BV administration included fatigue, nausea, fever, peripheral neuropathy, and cytopenias, which were consistent with the known safety profile of BV. Although it was hypothesized that BV might suppress cellular immunity by depleting CD30-positive activated T cells, no significant increase in the risk of cytomegalovirus antigenemia or infections was observed following BV administration [67,68]. Nevertheless, close monitoring for infectious complications remains warranted.

Additionally, BV may have the potential to mitigate GVHD by targeting CD30-positive T cells [70]. In the study by Tsirigotis et al., BV was administered in combination with DLI to some patients with relapsed Hodgkin lymphoma after transplantation. Although GVHD developed in 7 of the 10 patients following DLI, all five patients who required treatment achieved resolution of GVHD after a short course of low-dose steroids. Among the 6 patients who received BV alone, none developed acute or chronic GVHD. Their observations raise the possibility that BV-mediated depletion of alloreactive CD30-positive donor T cells may contribute to a reduction in both the incidence and severity of GVHD [69].

## 3. Lenalidomide (LEN)

LEN, a thalidomide analogue, is thought to exert its anti-tumor effects through multiple mechanisms. These include direct cytotoxicity against tumor cells [71]; promotion of T cell and natural killer cell proliferation and activation [72]; enhancement of antitumor immune responses by increasing the production of cytokines such as interleukin-2 (IL-2) and interferon-gamma (IFN-γ) [73]; suppression of Treg function [74]; inhibition of angiogenic factors such as vascular endothelial growth factor (VEGF), thereby preventing tumor growth and metastasis [75]; and binding to cereblon, leading to the degradation of lymphoid transcription factors IKZF1 and IKZF3 [76,77].

Although the clinical application of LEN initially focused on multiple myeloma, it has been investigated for various hematologic malignancies due to its multiple anti-tumor mechanisms. Regarding its application to T-cell lymphomas, several clinical trials have reported treatment outcomes with LEN monotherapy. The overall response rate has (ORR) been reported to be approximately 20–30% in PTCL [78,79,80] and 42% in ATL [21]. Although LEN demonstrated certain efficacy in some patients, both PFS and OS were relatively short, and its use is generally considered for selected patients who are expected to benefit from LEN. LEN is more commonly used as post-transplant maintenance therapy rather than as bridging therapy prior to transplantation. Therefore, the subsequent section discusses the impact of LEN administration after allo-HCT.

### Impact of Post-HCT Use

LEN administration early after allo-HCT has been associated with a high risk of severe GVHD [81,82], due to rapid proliferation of donor-derived T cells [83]. Delaying initiation until ≥3–6 months post-transplant and/or dose reduction has been reported to mitigate this risk [84,85,86,87]. Moreover, combining LEN with azacitidine (AZA), which can expand Tregs, may enhance antitumor activity without increasing GVHD incidence [88,89]. Nonetheless, cytopenia remains a major limitation, requiring careful dose and schedule adjustment.

Several case reports have described LEN administration for TCLs after allo-HCT, particularly in ATL [53,54,90]. In the largest published case series (*n* = 11), the median interval from allo-HCT to the initiation of LEN was 162 days (range: 43–1560 days), and the median initial daily dose of LEN was 10 mg (range: 5–25 mg) [91]. In this series, 3 patients (27%) achieved a CR, and 2 patients (18%) achieved a PR. The response rates were 57% for skin lesions and 50% for lymph node or extranodal lesions, which contrasts with findings observed with MOG [52]. GVHD was newly developed in 5 patients (45%) and worsened in 4 patients (36%), but in all cases, it was manageable. Interestingly, patients with a prior history of MOG treatment for relapse after allo-HCT showed better response rates to LEN compared to those without prior MOG exposure, suggesting that pre-treatment with MOG may enhance the therapeutic efficacy of LEN.

## 4. HDAC Inhibitors (Vorinostat, Romidepsin, Belinostat, Chidamide (Tucidinostat))

TCLs are characterized by the accumulation of genetic and epigenetic abnormalities through a multistep process [92]. Aberrations in signaling pathways such as T-cell receptor (TCR)/CD3, Notch, Janus kinase/signal transducer and activator of transcription (JAK/STAT), and Ras homolog family member A (RHOA), as well as mutations in epigenetic regulators including ten-eleven translocation 2 (TET2), DNA methyltransferase 3A (DNMT3A), and isocitrate dehydrogenase 2 (IDH2), have been implicated in their pathogenesis [93]. In addition to these intrinsic alterations, interactions with the tumor microenvironment promote tumor growth, survival, and progression [92]. Dysregulation of acetylation and deacetylation is known to induce genomic instability and aberrant gene expression in various malignancies, including TCLs. Histone deacetylases (HDACs) regulate downstream gene networks by deacetylating transcription factors and signaling mediators [94]. In TCLs, aberrant HDAC activity leads to a combination of pathogenic processes, including the silencing of tumor suppressor genes, inhibition of apoptotic pathways, promotion of the degradation of epigenetic regulators, enhancement of inflammatory signaling, and stimulation of angiogenesis [93,94]. Specifically, HDAC1/2/3 suppress the expression of STAT3 target genes, contributing to the epigenetic silencing of tumor suppressor genes [95]. HDAC1 also deacetylates tumor protein p53 (p53), resulting in diminished apoptotic signaling through reduced cyclin-dependent kinase inhibitor 1A (p21) expression [95]. Furthermore, HDAC1/2-mediated deacetylation and subsequent degradation of TET2 disrupt epigenetic regulation [96]. HDAC3 promotes nuclear factor kappa B (NF-κB) activation via the TCR/CD3 pathway, supporting tumor cell survival and the production of inflammatory cytokines [96]. In addition, HDAC activity maintains the expression of pro-angiogenic factors such as vascular endothelial growth factor (VEGF) and basic fibroblast growth factor (bFGF), facilitating tumor growth and progression [97]. Collectively, these abnormalities play a critical role in the initiation and malignant evolution of TCLs.

By modulating the above molecular and epigenetic pathways implicated in TCLs, HDAC inhibitors demonstrate antitumor efficacy. Several histone deacetylase inhibitors (HDACis) have been approved for the treatment of PTCL and CTCL. Vorinostat is approved by the U.S. FDA and the European Medicines Agency (EMA) for relapsed or refractory CTCL, with an ORR of approximately 30% [22,98]. Romidepsin has received FDA approval for both PTCL and CTCL, with reported ORRs of 25% in PTCL and 34% in CTCL [23,99]. Belinostat is approved by the FDA for relapsed or refractory PTCL, showing an ORR of 25% [24]. In China, the oral HDACi chidamide (also known internationally as tucidinostat) is approved for relapsed or refractory TCLs, with ORRs of 28% [100]. In addition, clinical outcomes of combination therapies incorporating HDACis with conventional chemotherapy, hypomethylating agents, proteasome inhibitors, or LEN have been reported, leading to an expansion of HDACi-based treatment strategies for TCLs [101]. For example, the efficacy of combination therapy with romidepsin and azacitidine in patients with PTCL has been reported, demonstrating particularly high activity in patients with a T-follicular helper cell phenotype (overall response rate [ORR] 80%, CR 50%) and in those harboring TET2 mutations (ORR 69%, CR 53%) [102].

### Control of GVHD by HDACis

The diverse epigenetic regulatory functions of HDACis have also been reported to be effective in modulating GVHD in the setting of allo-HCT (Figure 3) [103]. One of their key actions is the suppression of proinflammatory cytokine production. For example, vorinostat has been reported to inhibit the secretion and gene expression of cytokines such as IFN-γ and TNF-α at low concentrations [104]. HDACis also suppress the maturation and cytokine production of antigen-presenting cells (APCs), particularly dendritic cells, thereby preventing excessive immune activation [105]. Another important effect of HDACis is the enhancement of Tregs in terms of both number and function. This is mediated through the acetylation of Foxp3, which contributes to the stability and suppressive capacity of Tregs [106]. Clinical studies have demonstrated that vorinostat administration is associated with enhanced Treg-mediated immune regulation [107]. In addition, HDACis have been shown to influence the activity of natural killer (NK) cells. Entinostat, for instance, enhances the expression of NK cell activation markers and promotes cytotoxic function via specific signaling pathways, suggesting a potential role in GVHD suppression through NK cell–mediated mechanisms [108]. Furthermore, HDACis also modulate the balance of helper T-cell subsets involved in GVHD pathogenesis [109]. Taken together, these findings indicate that HDACis may suppress the development and progression of GVHD through multiple pathways, including the regulation of inflammatory cytokines and modulation of APCs, Tregs, NK cells, and helper T-cell subsets. These agents are thus emerging as promising options for post-transplant immune modulation.

Although no trials have specifically targeted TCLs, several studies have investigated the use of HDACis in combination with allo-HCT. In a phase 1/2 study reported by Choi et al., vorinostat (either 100 mg or 200 mg, twice a day) from Day -10 to Day 100 was administered in combination with tacrolimus (TAC) and mycophenolate mofetil (MMF) for GVHD prophylaxis following reduced-intensity conditioning and transplantation from HLA-matched donors. The cumulative incidence of grade 2–4 acute GVHD was relatively low at 22%, and the addition of vorinostat was considered safe [110]. Furthermore, they also evaluated a similar GVHD prophylaxis strategy using vorinostat in the setting of myeloablative conditioning with unrelated donor transplantation, reporting a favorable cumulative incidence of grade 2–4 acute GVHD at 22% [107]. Bug et al. evaluated the safety of post-transplant maintenance therapy with panobinostat in patients with high-risk acute myeloid leukemia (AML) and myelodysplastic syndromes (MDS) [111]. Similarly, Kalin et al. investigated a treatment strategy in which panobinostat and decitabine were administered early after transplantation, followed by donor lymphocyte infusion (DLI), in patients with high-risk AML/MDS [112]. Despite the inclusion of high-risk populations in both studies, the relapse rates were relatively low, and the administration of panobinostat was concluded to be safe.

## 5. EZH1/2 Inhibitors

Polycomb group (PcG) proteins are epigenetic regulators that control the expression of target genes through the modulation of histone modifications. Mutations and overexpression of enhancer of zeste homolog 1 and 2 (EZH1/2), histone methyltransferases that are core components of the polycomb repressive complex (PRC), have been reported in various hematologic malignancies [113]. The first selective EZH2 inhibitor (tazemetostat) was developed, which demonstrated antitumor activity in B-cell non-Hodgkin lymphoma cell lines and was approved for EZH2-mutated follicular lymphoma [114]. Moreover, EZH2 inhibition alone is insufficient in some tumors and is ineffective in tumors lacking EZH2 expression [115].

Against this background, the development of dual EZH1/2 inhibitors, which simultaneously target both EZH1 and EZH2, has progressed to overcome the limitations of EZH2-selective inhibition (Figure 4) [115]. Valemetostat is a representative agent in this class, and by inhibiting both EZH1 and EZH2, it is expected to achieve broader epigenetic reprogramming and enhanced antitumor efficacy [116]. Valemetostat has demonstrated promising results in clinical trials involving patients with relapsed or refractory PTCL and ATL, and was approved in Japan in 2022 for the treatment of these malignancies [117,118]. In particular, high response rates and favorable tolerability have been reported in disease subtypes characterized by EZH2 overexpression or activation, such as T follicular helper (Tfh) cell–derived PTCL [118].

### Impact of EZH1/2 Inhibition on GVHD

Several murine models have demonstrated that EZH2 plays a critical role in the pathogenesis of GVHD [119]. For example, it has been reported that alloreactive effector T cells express high levels of Ezh2, and that genetic deletion of Ezh2 in donor T cells significantly alleviates GVHD in murine models of allo-HCT [120,121,122]. However, none of the studies have shown sufficient GVHD suppression with selective EZH2 inhibition alone, indicating the need for further preclinical and clinical investigations, including those involving dual EZH1/2 inhibitors. Clinical studies evaluating the use of EZH1/2 inhibitors in the pre- or post-transplant setting remain extremely limited. Bagnato et al. reported two cases of refractory/relapsed PTCL in which valemetostat was administered as bridging therapy prior to allo-HCT [123]. In both cases, valemetostat did not appear to have a negative impact on engraftment or the development of GVHD. Valemetostat is considered a promising therapeutic agent, particularly for ATL, and further clinical study is warranted to evaluate its safety and efficacy in the peri-transplant setting.

## 6. Immune Checkpoint Inhibitors (ICIs)

The activation of cytotoxic T cells requires not only signals from T cell receptors that recognize peptides presented on HLA molecules, but also co-stimulatory signals mediated by activating co-stimulatory molecules such as CD28. In contrast, some co-stimulatory molecules, such as CTLA-4 and PD-1, function as inhibitory molecules and suppress T cell activation. The pathways mediated by these inhibitory co-stimulatory molecules are known as immune checkpoints and play a critical role in tumor immune evasion [124]. Antibody drugs targeting these immune checkpoints, known as immune checkpoint inhibitors (ICIs), have been developed and have come into widespread use over the past decade for various treatment-resistant cancers [125].

Clinical trials are currently underway for various subtypes of TCLs [126]. Notably, promising treatment outcomes have been reported for extranodal NK/T-cell lymphoma, particularly with the use of pembrolizumab as salvage therapy in relapsed or refractory cases, with several studies demonstrating high response rates and durable remissions in a subset of patients [27,127]. On the other hand, there have been reports of rapid disease progression following nivolumab administration in indolent ATL, suggesting that caution may be needed when using ICIs in certain types of TCLs [128].

### Impact of Pre- and Post-HCT Use

Owing to their demonstrated efficacy as salvage therapy, ICIs are anticipated to serve as an effective bridging strategy prior to allo-HCT. However, based on reports focusing on Hodgkin lymphoma, the use of ICIs before allo-HCT is associated with an increased risk of GVHD [129,130,131,132]. Immune profiling suggested persistent immune alterations, including reduced PD-1+ T cells and lower Treg-to-effector T cell ratios [133]. Interestingly, several studies have reported that the increased risk of GVHD associated with the use of ICIs prior to allo-HCT may be mitigated by the use of PTCy as GVHD prophylaxis [134,135]. In allo-HCT for patients previously treated with ICIs, careful optimization of GVHD prophylaxis is of critical importance. On the other hand, the use of ICIs after allo-HCT has the potential to mitigate tumor immune evasion and T-cell exhaustion, thereby enhancing the GVL effect [136]. However, in such cases, careful monitoring is warranted not only for GVHD but also for immune-related adverse events associated with ICIs.

## 7. Conclusions

We have summarized the effects of the novel agents discussed so far on allo-HCT in Table 1. The integration of novel agents into the management of TCLs has provided new therapeutic opportunities, both before and after allo-HCT. These agents have not only anti-tumor activity but also immune-modulating effects; we should pay attention to how, for what purpose, in which patients, and at what timing they should be combined with allo-HCT. Unfortunately, TCLs are rare malignancies, and the number of patients undergoing transplantation remains limited. Consequently, data on the integration of novel agents with allo-HCT in TCLs are scarce, and robust evidence has yet to be established. Furthermore, novel T-cell and NK-cell immunotherapies—such as bispecific antibodies, chimeric antigen receptor (CAR)-engineered T cells, or CAR-engineered NK cells targeting CD7 [137], CD30 [138,139,140], CD70 [141], and T-cell receptor beta constant region 1 (TRBC1) [142,143]—are currently under active development for TCLs. These advances have the potential to substantially reshape the role of allo-HCT. Further clinical and translational studies are warranted to establish the optimal strategies for combining these therapies in this setting.

## Figures and Tables

**Figure 1 cells-14-01306-f001:**
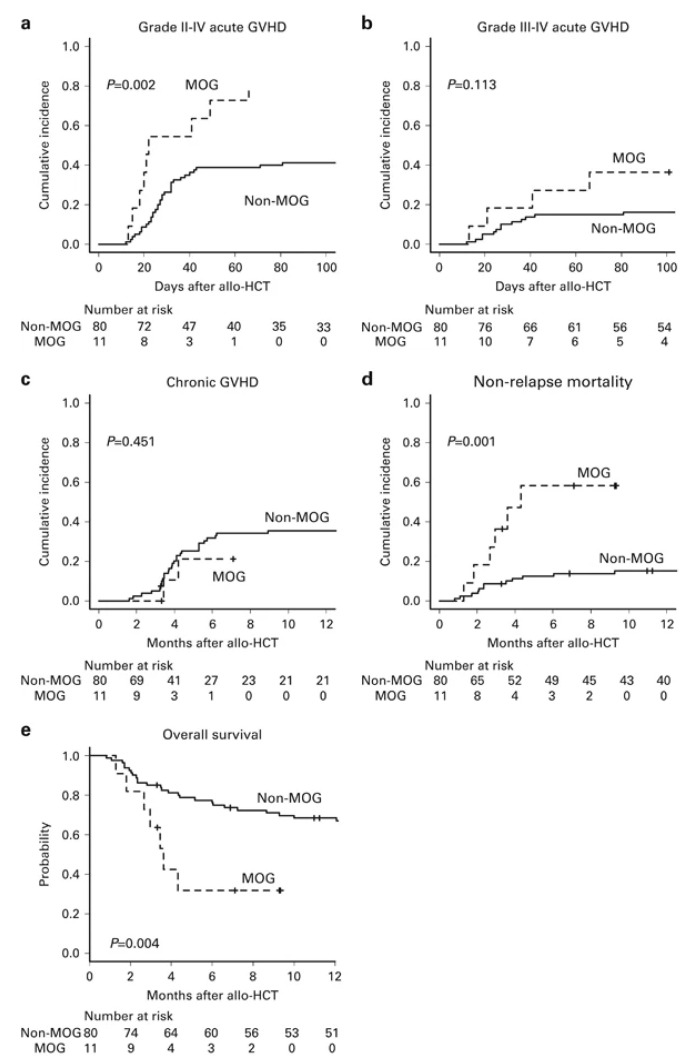
Impact of pretransplantation mogamulizumab on transplant outcomes. The cumulative incidence of grade II–IV acute GVHD: (**a**). grade III–IV acute GVHD (**b**). chronic GVHD (**c**). and non-relapse mortality (**d**). The probability of OS (**e**). allo-HCT, allogeneic hematopoietic cell transplantation; MOG, mogamulizumab [38].

**Figure 2 cells-14-01306-f002:**
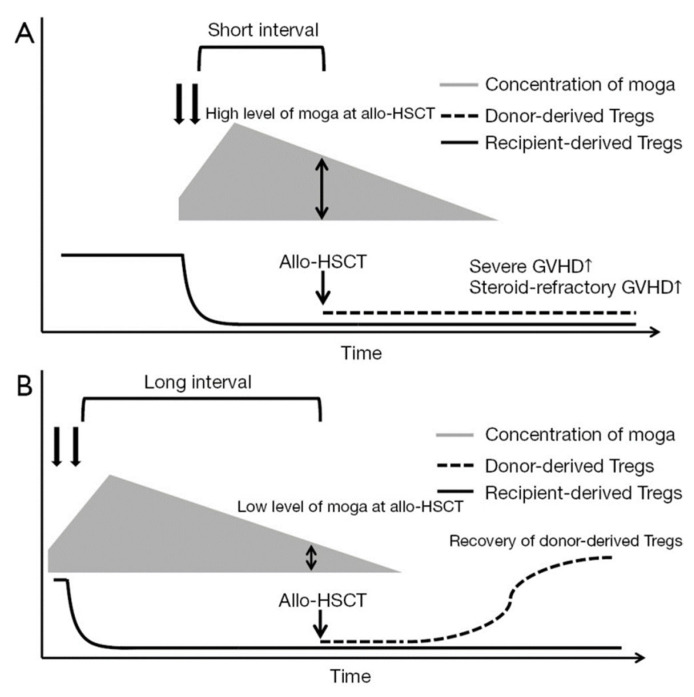
Importance of the interval between the administration of moga and allo-HSCT. (**A**). With a short interval between moga administration and allo-HSCT, the concentration of moga at allo-HSCT might be high, which is sufficient to deplete donor-derived Tregs as well as recipient-derived Tregs; (**B**). With a long interval between moga administration and allo-HSCT, the concentration of moga at allo-HSCT might be low, which is no longer able to deplete donor-derived Tregs, although moga has depleted recipient-derived Tregs [43].

**Figure 3 cells-14-01306-f003:**
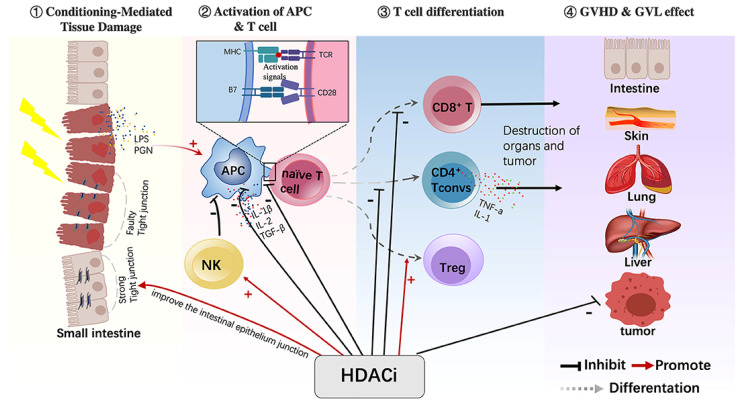
Immunomodulatory Actions of HDAC Inhibitors in the Suppression of GVHD. HDACis have various immunomodulatory effects on different cells. HDACis play an important role in regulating the maturation of APCs, reducing their antigen-presenting capacity and inhibiting the production of proinflammatory cytokines. HDACis also promote the conversion of naive T cells into Tregs and increase their function. In addition, HDACis activate NK cells and inhibit CD4+ Tconv cells and CD8+ T cells. Moreover, HDACis can improve the intestinal epithelium junction during the GVHD process (HDACi, histone deacetylase inhibitor; APC, antigen-presenting cell; Treg, regulatory T cell; NK, natural killer cell; CD4+ Tconv, CD4+ conventional T cells) [103].

**Figure 4 cells-14-01306-f004:**
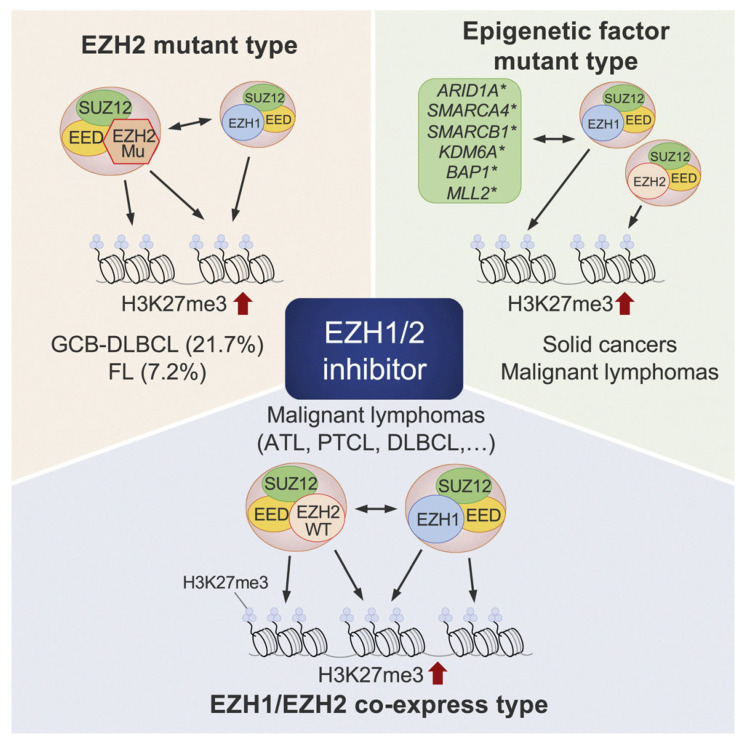
EZH1/2 Co-Expression and H3K27me3 Dysregulation in T-Cell Lymphomas. Increased histone H3 modification (particularly H3K27me3) by EZH1 or EZH2 has been reported to contribute to the progression of various cancers. In TCLs such as ATL and PTCL, co-expression of EZH1 and EZH2 leads to elevated levels of H3K27me3, contributing to tumorigenesis. Valemetostat, which inhibits not only EZH2 but also EZH1, is expected to be an effective therapeutic agent for TCLs. (ATL; Adult T-cell leukemia/lymphoma, EZH1/2; Enhancer of zeste homolog 1/2, H3K27me3; Histone H3 lysine 27 trimethylation, PTCL; Peripheral T-cell lymphoma, TCLs; T-cell lymphomas) [115].

**Table 1 cells-14-01306-t001:** Summary of the Impact of Salvage Therapies on T-Cell Lymphomas Pre- and Post-Allogeneic Hematopoietic Cell Transplantation.

Novel Agents for TCLs		Effects on HCT Immunity	Pre-HCT Use	Post-HCT Use
Conventional salvage chemotherapy		Non-specific cytotoxic effects on both malignant and normal hematopoietic/immune cells; potential impairment of immune reconstitution.	Commonly used as salvage therapy to achieve disease control; responses are often short-lived in chemotherapy-refractory TCLs.	Limited role due to myelosuppression and cumulative toxicity; generally avoided except in selected relapse cases.
Antibody drugs	MOG	Depletes Tregs.	Increase the risk of steroid-refractory GVHD and NRM (especially with a short MOG-to-HCT interval).	∙Relatively safe ≥ 3 months post-HCT. ∙Effective against ATL peripheral blood lesions.
	BV	Depletes activated CD30+ T cells.	Safe as bridging therapy without affecting engraftment or GVHD.	May help reduce GVHD.
LEN		∙Activates T and NK cells. ∙Increases cytokine production. ∙Suppresses Treg function.	Data limited.	∙Early use after HCT increases the risk of GVHD. ∙Delayed use or combination with AZA may reduce GVHD risk. ∙Myelosuppression is a concern.
HDACis		∙Suppress cytokine production. ∙Stabilize Tregs ∙Modulate APC and NK cell function.	Data limited.	Potential for GVHD prevention; well tolerated.
EZH1/2 inhibitors		May modulate GVHD via epigenetics.	Limited reports; no clear harm observed.	No established role; further studies needed.
Immune checkpoint inhibitors		Reduced PD-1+ T cells and Tregs.	Increased risk of GVHD; however, this may be mitigated by the use of PTCy as GVHD prophylaxis.	∙May enhance the GVL effect. ∙Requires monitoring for both GVHD and irAEs.

## Data Availability

No new data were created or analyzed in this study. Data sharing is not applicable to this article.

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
