# Peer review of "Impact of Novel Agents on Allogeneic Hematopoietic Cell Transplantation in Patients with T-Cell Lymphomas"

_cells, 2025, doi:10.3390/cells14171306_

Round 1

Reviewer 1 Report

Comments and Suggestions for Authors

The authors summarize the available data for the use of novel agents: monoclonal Abs, DACs, IMIDs and epigenetic targeting therapies in T-cell lymphomas, undergoing allogeneic BMT. The manuscript is well written and could benefit from some suggested changes.

 1.  The aim is to present data for mature T-cell lymphomas, however cutaneous T-cell lymphomas are not covered, where is significant prospective and retrospective data for the benefit of alloBMT, especially in subtypes such as Sezary syndrome, which appear to have high cure rates. These lymphomas are approved indications for treatment with MOG and HDACis, therefore it would be important to include them in the review.

Also, the PTCL subtype of AITL/ Tfh is not adequately covered in the review. This entity appears to have relatively better outcomes after alloBMT as compared to other types of PTCL. Importantly, lymphomas with RHOA, DNMT3A and TET2 mutations have very high rates of response to epigenetic therapies with ORR of 70% and CR of 50% to the combination of romidepsin and azacitidine.

2. The importance of disease responsive to salvage therapy pre alloBMT should be discussed as PD/ refractory disease negatively impacts outcomes in most analyses, especially the recent large CIBMTR and SFGM-TC cohorts This raises the question of the best salvage regimen pre allograft, and a brief comparison of the available novel agents to standard common salvage chemotherapy regimens would be useful.

3. The section on lenalidomide mainly discusses data around other haematologic cancers and has low relevance to the majority of T-cell lymphomas and could be shortened significantly.

4. While there appears to be a signal for the risks of significant GVHD for MOG use peritransplant, it is important to highlight that the recommendations are based on retrospective analysis of small datasets. 

Similar considerations apply to the use of immune checkpoint inhibitors. The role of pembrolizumab as salvage therapy prior to alloBMT in ENKTL has not been mentioned in the review.

5. In future directions it would be useful to briefly mention newer novel agents such as T-cell/ NK cell redirecting therapies using bsAbs/ CART cells/ CAR NK cells targeting CD30, CD70 etc. Early reports suggest the potential for significant efficacy, and the possibility for use as a bridge to alloBMT.

Author Response

Reviewer: 1

The authors summarize the available data for the use of novel agents: monoclonal Abs, DACs, IMIDs and epigenetic targeting therapies in T-cell lymphomas, undergoing allogeneic BMT. The manuscript is well written and could benefit from some suggested changes.

Response:

We are grateful for your positive assessment of our review.

  1. The aim is to present data for mature T-cell lymphomas, however cutaneous T-cell lymphomas are not covered, where is significant prospective and retrospective data for the benefit of alloBMT, especially in subtypes such as Sezary syndrome, which appear to have high cure rates. These lymphomas are approved indications for treatment with MOG and HDACis, therefore it would be important to include them in the review.

Response:

Thank you for your valuable comment. In our review, we intended to address the relationship between novel agents and allo-HCT in T-cell lymphomas, including cutaneous T-cell lymphomas (CTCL). However, we recognize that this was not clearly stated, which may have led to misunderstanding. We also acknowledge that the discussion on the benefits of allo-HCT for CTCL and the approved indications for MOG and HDAC inhibitors was insufficient. In response to the reviewer’s comment, we have revised or added the contents as follows:

Page 2, Line 33,

T-cell lymphomas (TCLs) are a heterogeneous group of non-Hodgkin lymphomas (NHLs) that originate from mature T cells or natural killer (NK) cells, and include both systemic/peripheral T-cell and NK/T-cell lymphomas (PTCL/NKTCL) and primary cutaneous T-cell lymphomas (CTCL).

Page 2, Line 40,

… adult T-cell leukemia/lymphoma (ATL), as well as mycosis fungoides (MF) and Sézary syndrome (SS) among cutaneous types, each with distinct clinical and pathological features [4,5].

Page 2, Line 52,

For example, in advanced-stage MF and SS, multiple prospective and retrospective studies have demonstrated the efficacy of allo-HCT, with 3-year overall survival rates reaching 40–60% [12-14].

Page 4, Line 85,

In the United States, it was approved by the Food and Drug Administration (FDA) in 2018 following a phase III clinical trial targeting CTCL, specifically MF and SS [35].

We have added the following references in conjunction with the revision.

  1. Dobos, G.; Pohrt, A.; Ram-Wolff, C.; Lebbe, C.; Bouaziz, J.D.; Battistella, M.; Bagot, M.; de Masson, A. Epidemiology of Cutaneous T-Cell Lymphomas: A Systematic Review and Meta-Analysis of 16,953 Patients. Cancers (Basel) 2020, 12, doi:10.3390/cancers12102921.

  1. Weng, W.K.; Arai, S.; Rezvani, A.; Johnston, L.; Lowsky, R.; Miklos, D.; Shizuru, J.; Muffly, L.; Meyer, E.; Negrin, R.S.; et al. Nonmyeloablative allogeneic transplantation achieves clinical and molecular remission in cutaneous T-cell lymphoma. Blood Adv 2020, 4, 4474-4482, doi:10.1182/bloodadvances.2020001627.
  2. Domingo-Domenech, E.; Duarte, R.F.; Boumedil, A.; Onida, F.; Gabriel, I.; Finel, H.; Arcese, W.; Browne, P.; Beelen, D.; Kobbe, G.; et al. Allogeneic hematopoietic stem cell transplantation for advanced mycosis fungoides and Sézary syndrome. An updated experience of the Lymphoma Working Party of the European Society for Blood and Marrow Transplantation. Bone Marrow Transplant 2021, 56, 1391-1401, doi:10.1038/s41409-020-01197-3.
  3. Goyal, A.; O'Leary, D.; Foss, F. Allogeneic stem cell transplant for treatment of mycosis fungoides and Sezary syndrome: a systematic review and meta-analysis. Bone Marrow Transplant 2024, 59, 41-51, doi:10.1038/s41409-023-02122-0.

Also, the PTCL subtype of AITL/ Tfh is not adequately covered in the review. This entity appears to have relatively better outcomes after alloBMT as compared to other types of PTCL. Importantly, lymphomas with RHOA, DNMT3A and TET2 mutations have very high rates of response to epigenetic therapies with ORR of 70% and CR of 50% to the combination of romidepsin and azacitidine.

Response:

In addition, we acknowledge that our discussion on AITL/Tfh was also insufficient. We have therefore added content on AITL/Tfh, incorporating the points you suggested, as follows:

Page 3, Line 54,

In AITL, an analysis of 249 patients with refractory disease or relapse after autologous transplantation reported 4-year PFS and OS rates of 49% and 56%, respectively [15].

Page 16, Line 287,

For example, the efficacy of combination therapy with romidepsin and azacitidine in patients with PTCL has been reported, demonstrating particularly high activity in patients with a T-follicular helper cell phenotype (overall response rate [ORR] 80%, CR 50%) and in those harboring TET2 mutations (ORR 69%, CR 53%) [102].

We have added the following references in conjunction with the revision.

  1. Epperla, N.; Ahn, K.W.; Litovich, C.; Ahmed, S.; Battiwalla, M.; Cohen, J.B.; Dahi, P.; Farhadfar, N.; Farooq, U.; Freytes, C.O.; et al. Allogeneic hematopoietic cell transplantation provides effective salvage despite refractory disease or failed prior autologous transplant in angioimmunoblastic T-cell lymphoma: a CIBMTR analysis. J Hematol Oncol 2019, 12, 6, doi:10.1186/s13045-018-0696-z.

  1. Falchi, L.; Ma, H.; Klein, S.; Lue, J.K.; Montanari, F.; Marchi, E.; Deng, C.; Kim, H.A.; Rada, A.; Jacob, A.T.; et al. Combined oral 5-azacytidine and romidepsin are highly effective in patients with PTCL: a multicenter phase 2 study. Blood 2021, 137, 2161-2170, doi:10.1182/blood.2020009004.

  1. The importance of disease responsive to salvage therapy pre alloBMT should be discussed as PD/ refractory disease negatively impacts outcomes in most analyses, especially the recent large CIBMTR and SFGM-TC cohorts This raises the question of the best salvage regimen pre allograft, and a brief comparison of the available novel agents to standard common salvage chemotherapy regimens would be useful.

Response:

Thank you for your insightful comment. We have incorporated data from the CIBMTR and SFGM-TC cohorts and added the following description accordingly. In addition, we have updated Table 1 to include the impact of standard salvage chemotherapy administered before and after allo-HCT on transplant outcomes.

Page 3, Line 65,

In the context of transplantation for TCLs, achieving optimal disease control prior to allo-HCT is one of the most critical factors of outcome [15,28]. Given the activity of these novel agents in TCLs, they are increasingly being utilized as salvage therapy for chemotherapy-refractory disease or as a bridge to allo-HCT. Furthermore, unlike conventional cytotoxic agents, some of these drugs have relatively limited myelosuppression, making them feasible for use as post-transplant maintenance therapy aimed at reducing the risk of relapse.

We have added the following references in conjunction with the revision.

  1. Mamez, A.C.; Dupont, A.; Blaise, D.; Chevallier, P.; Forcade, E.; Ceballos, P.; Mohty, M.; Suarez, F.; Beguin, Y.; Peffault De Latour, R.; et al. Allogeneic stem cell transplantation for peripheral T cell lymphomas: a retrospective study in 285 patients from the Societe Francophone de Greffe de Moelle et de Therapie Cellulaire (SFGM-TC). J Hematol Oncol 2020, 13, 56, doi:10.1186/s13045-020-00892-4.

  1. The section on lenalidomide mainly discusses data around other haematologic cancers and has low relevance to the majority of T-cell lymphomas and could be shortened significantly.

Response:

Thank you for your comment. In accordance with your suggestion, we have substantially removed data unrelated to TCLs regarding the impact of pre-transplant LEN administration and have revised the section to present a more concise summary, as follows:

Page 13, Line 240,

LEN administration early after allo-HCT has been associated with a high risk of severe GVHD [81,82], due to rapid proliferation of donor-derived T cells [83]. Delaying initiation until ≥3–6 months post-transplant and/or dose reduction has been reported to mitigate this risk [84-87]. Moreover, combining LEN with azacitidine (AZA), which can expand Tregs, may enhance anti-tumor activity without increasing GVHD incidence [88,89]. Nonetheless, cytopenia remain a major limitation, requiring careful dose and schedule adjustment.

  1. While there appears to be a signal for the risks of significant GVHD for MOG use peritransplant, it is important to highlight that the recommendations are based on retrospective analysis of small datasets.

Response:

Thank you for your comment. As you pointed out, the evidence regarding the impact of MOG administration, particularly in the post-HCT setting, is limited to small retrospective analyses. In accordance with your suggestion, we have added the following statement to clarify this point.

Page 10, Line 168,

However, reports on the use of MOG after allo-HCT are limited to small retrospective series, and further investigations from both basic and clinical perspectives are required.

Similar considerations apply to the use of immune checkpoint inhibitors. The role of pembrolizumab as salvage therapy prior to alloBMT in ENKTL has not been mentioned in the review.

Response:

Thank you very much for this thoughtful comment. We also received similar feedback from another reviewer regarding immune checkpoint inhibitors, and in response, we have added a new section on immune checkpoint inhibitors as outlined below.

Page 21, Line 382,

Immune checkpoint inhibitors (ICIs)

The activation of cytotoxic T cells requires not only signals from T cell receptors that recognize peptides presented on HLA molecules, but also co-stimulatory signals mediated by activating co-stimulatory molecules such as CD28. In contrast, some co-stimulatory molecules, such as CTLA-4 and PD-1, function as inhibitory molecules and suppress T cell activation. The pathways mediated by these inhibitory co-stimulatory molecules are known as immune checkpoints and play a critical role in tumor immune evasion [124]. Antibody drugs targeting these immune checkpoints, known as immune checkpoint inhibitors (ICIs), have been developed and have come into widespread use over the past decade for various treatment-resistant cancers [125].

Clinical trials are currently underway for various subtypes of TCLs [126]. Notably, promising treatment outcomes have been reported for extranodal NK/T-cell lymphoma, particularly with the use of pembrolizumab as salvage therapy in relapsed or refractory cases, with several studies demonstrating high response rates and durable remissions in a subset of patients [27,127]. On the other hand, there have been reports of rapid disease progression following nivolumab administration in indolent ATL, suggesting that caution may be needed when using ICIs in certain types of TCLs [128].

Impact of pre- and post-HCT use

Owing to their demonstrated efficacy as salvage therapy, ICIs are anticipated to serve as an effective bridging strategy prior to allo-HCT. However, based on reports focusing on Hodgkin lymphoma, the use of ICIs before allo-HCT is associated with an increased risk of GVHD [129-132]. Immune profiling suggested persistent immune alterations, including reduced PD-1+ T cells and lower Treg-to-effector T cell ratios [133]. Interestingly, several studies have reported that the increased risk of GVHD associated with the use of ICIs prior to allo-HCT may be mitigated by the use of PTCy as GVHD prophylaxis [134,135]. In allo-HCT for patients previously treated with ICIs, careful optimization of GVHD prophylaxis is of critical importance. On the other hand, the use of ICIs after allo-HCT has the potential to mitigate tumor immune evasion and T-cell exhaustion, thereby enhancing the GVL effect [136]. However, in such cases, careful monitoring is warranted not only for GVHD but also for immune-related adverse events associated with ICIs.

In addition, corresponding updates have been made to the abstract, introduction, and Table 1 to incorporate this information.

Page 1, Line 26 (Abstract),

… histone deacetylase inhibitors, enhancer of zeste homolog inhibitors, and immune checkpoint inhibitors—before or after allo-HCT…

Page 3, Line 64,

… enhancer of zeste homolog 1/2 (EZH1/2) inhibitors (e.g., valemetostat [26]), and immune checkpoint inhibitors (ICIs) (e.g., pembrolizumab [27]).

We have added the following references in conjunction with the revision.

  1. Kwong, Y.L.; Chan, T.S.Y.; Tan, D.; Kim, S.J.; Poon, L.M.; Mow, B.; Khong, P.L.; Loong, F.; Au-Yeung, R.; Iqbal, J.; et al. PD1 blockade with pembrolizumab is highly effective in relapsed or refractory NK/T-cell lymphoma failing l-asparaginase. Blood 2017, 129, 2437-2442, doi:10.1182/blood-2016-12-756841.

  1. Chen, L.; Flies, D.B. Molecular mechanisms of T cell co-stimulation and co-inhibition. Nature reviews. Immunology 2013, 13, 227-242, doi:10.1038/nri3405.

  1. Mc Neil, V.; Lee, S.W. Advancing Cancer Treatment: A Review of Immune Checkpoint Inhibitors and Combination Strategies. Cancers 2025, 17, 1408.

  1. Chen, X.; Wu, W.; Wei, W.; Zou, L. Immune Checkpoint Inhibitors in Peripheral T-Cell Lymphoma. Front Pharmacol 2022, 13, 869488, doi:10.3389/fphar.2022.869488.

  1. Li, X.; Cheng, Y.; Zhang, M.; Yan, J.; Li, L.; Fu, X.; Zhang, X.; Chang, Y.; Sun, Z.; Yu, H.; et al. Activity of pembrolizumab in relapsed/refractory NK/T-cell lymphoma. J Hematol Oncol 2018, 11, 15, doi:10.1186/s13045-018-0559-7.

  1. Ratner, L.; Waldmann, T.A.; Janakiram, M.; Brammer, J.E. Rapid Progression of Adult T-Cell Leukemia-Lymphoma after PD-1 Inhibitor Therapy. N Engl J Med 2018, 378, 1947-1948, doi:10.1056/NEJMc1803181.

  1. Casadei, B.; Broccoli, A.; Stefoni, V.; Pellegrini, C.; Marangon, M.; Morigi, A.; Nanni, L.; Lolli, G.; Carella, M.; Argnani, L.; et al. PD-1 blockade as bridge to allogeneic stem cell transplantation in relapsed/refractory Hodgkin lymphoma patients: a retrospective single center case series. Haematologica 2019, 104, e521-e522, doi:10.3324/haematol.2019.215962.

  1. Ijaz, A.; Khan, A.Y.; Malik, S.U.; Faridi, W.; Fraz, M.A.; Usman, M.; Tariq, M.J.; Durer, S.; Durer, C.; Russ, A.; et al. Significant Risk of Graft-versus-Host Disease with Exposure to Checkpoint Inhibitors before and after Allogeneic Transplantation. Biol Blood Marrow Transplant 2019, 25, 94-99, doi:10.1016/j.bbmt.2018.08.028.

  1. Soiffer, R.J. Checkpoint inhibition to prevent or treat relapse in allogeneic hematopoietic cell transplantation. Bone Marrow Transplant 2019, 54, 798-802, doi:10.1038/s41409-019-0617-y.

  1. Bobillo, S.; Nieto, J.C.; Barba, P. Use of checkpoint inhibitors in patients with lymphoid malignancies receiving allogeneic cell transplantation: a review. Bone Marrow Transplant 2021, 56, 1784-1793, doi:10.1038/s41409-021-01268-z.

  1. Merryman, R.W.; Kim, H.T.; Zinzani, P.L.; Carlo-Stella, C.; Ansell, S.M.; Perales, M.A.; Avigdor, A.; Halwani, A.S.; Houot, R.; Marchand, T.; et al. Safety and efficacy of allogeneic hematopoietic stem cell transplant after PD-1 blockade in relapsed/refractory lymphoma. Blood 2017, 129, 1380-1388, doi:10.1182/blood-2016-09-738385.

  1. Merryman, R.W.; Castagna, L.; Giordano, L.; Ho, V.T.; Corradini, P.; Guidetti, A.; Casadei, B.; Bond, D.A.; Jaglowski, S.; Spinner, M.A.; et al. Allogeneic transplantation after PD-1 blockade for classic Hodgkin lymphoma. Leukemia 2021, 35, 2672-2683, doi:10.1038/s41375-021-01193-6.

  1. Hu, Y.; Wang, Y.; Min, K.; Zhou, H.; Gao, X. The influence of immune checkpoint blockade on the outcomes of allogeneic hematopoietic stem cell transplantation. Frontiers in immunology 2024, 15, 1491330, doi:10.3389/fimmu.2024.1491330.

  1. Norde, W.J.; Maas, F.; Hobo, W.; Korman, A.; Quigley, M.; Kester, M.G.; Hebeda, K.; Falkenburg, J.H.; Schaap, N.; de Witte, T.M.; et al. PD-1/PD-L1 interactions contribute to functional T-cell impairment in patients who relapse with cancer after allogeneic stem cell transplantation. Cancer research 2011, 71, 5111-5122, doi:10.1158/0008-5472.Can-11-0108.

  1. In future directions it would be useful to briefly mention newer novel agents such as T-cell/ NK cell redirecting therapies using bsAbs/ CART cells/ CAR NK cells targeting CD30, CD70 etc. Early reports suggest the potential for significant efficacy, and the possibility for use as a bridge to alloBMT.

Response:

Thank you for your insightful comment. We agree that this is a highly important point, and in response, we have added the following statements regarding bispecific antibodies and CAR- T and NK as a future direction.

Page 23, Line 417,

Furthermore, novel T-cell and NK-cell immunotherapies—such as bispecific antibodies, chimeric antigen receptor (CAR)-engineered T cells or CAR-engineered NK cells targeting CD7 [137], CD30 [138-140], CD70 [141], and T-cell receptor beta constant region 1 (TRBC1) [142,143]— are currently under active de-velopment for TCLs.

We have added the following references in conjunction with the revision.

  1. Hu, Y.; Zhang, M.; Yang, T.; Mo, Z.; Wei, G.; Jing, R.; Zhao, H.; Chen, R.; Zu, C.; Gu, T.; et al. Sequential CD7 CAR T-Cell Therapy and Allogeneic HSCT without GVHD Prophylaxis. N Engl J Med 2024, 390, 1467-1480, doi:10.1056/NEJMoa2313812.

  1. Moskowitz, A.; Harstrick, A.; Emig, M.; Overesch, A.; Pinto, S.; Ravenstijn, P.; Schlüter, T.; Rubel, J.; Rebscher, H.; Graefe, T.; et al. AFM13 in Combination with Allogeneic Natural Killer Cells (AB-101) in Relapsed or Refractory Hodgkin Lymphoma and CD30 + Peripheral T-Cell Lymphoma: A Phase 2 Study (LuminICE). Blood 2023, 142, 4855-4855, doi:10.1182/blood-2023-174250.

  1. Grover, N.S.; Hucks, G.; Riches, M.L.; Ivanova, A.; Moore, D.T.; Shea, T.C.; Seegars, M.B.; Armistead, P.M.; Kasow, K.A.; Beaven, A.W.; et al. Anti-CD30 CAR T cells as consolidation after autologous haematopoietic stem-cell transplantation in patients with high-risk CD30+ lymphoma: a phase 1 study. The Lancet Haematology 2024, 11, e358-e367, doi:https://doi.org/10.1016/S2352-3026(24)00064-4.

  1. Nieto, Y.; Banerjee, P.; Kaur, I.; Basar, R.; Li, Y.; Daher, M.; Rafei, H.; Kerbauy, L.N.; Kaplan, M.; Marin, D.; et al. Allogeneic NK cells with a bispecific innate cell engager in refractory relapsed lymphoma: a phase 1 trial. Nature Medicine 2025, 31, 1987-1993, doi:10.1038/s41591-025-03640-8.

  1. Iyer, S.P.; Sica, R.A.; Ho, P.J.; Prica, A.; Zain, J.; Foss, F.M.; Hu, B.; Beitinjaneh, A.; Weng, W.-K.; Kim, Y.H.; et al. Safety and activity of CTX130, a CD70-targeted allogeneic CRISPR-Cas9-engineered CAR T-cell therapy, in patients with relapsed or refractory T-cell malignancies (COBALT-LYM): a single-arm, open-label, phase 1, dose-escalation study. The Lancet Oncology 2025, 26, 110-122, doi:10.1016/S1470-2045(24)00508-4.

  1. Nichakawade, T.D.; Ge, J.; Mog, B.J.; Lee, B.S.; Pearlman, A.H.; Hwang, M.S.; DiNapoli, S.R.; Wyhs, N.; Marcou, N.; Glavaris, S.; et al. TRBC1-targeting antibody-drug conjugates for the treatment of T cell cancers. Nature 2024, 628, 416-423, doi:10.1038/s41586-024-07233-2.

  1. Cwynarski, K.; Iacoboni, G.; Tholouli, E.; Menne, T.; Irvine, D.A.; Balasubramaniam, N.; Wood, L.; Shang, J.; Xue, E.; Zhang, Y.; et al. TRBC1-CAR T cell therapy in peripheral T cell lymphoma: a phase 1/2 trial. Nat Med 2025, 31, 137-143, doi:10.1038/s41591-024-03326-7.

Reviewer 2 Report

Comments and Suggestions for Authors

Nice review. It might be improved as follows:

1- It is somewhat ATL- and Japan-centered, it would benefit from a more global TCL and geographical approach 

2- Including some reference to more standard agents would be helpful to clinicians (bexarotene, IFNa, methotrexate, ECP, CHOP, checkpoint inhibitors, etc), following Table 1 format

3- In the antibody drugs, some mention should be included of trials targeting other surface T-cell molecules (TRBC1, CD7, etc), such as PMID: 38538786/39528665, 38657244...

Author Response

Nice review. It might be improved as follows:

Response:

We are grateful for your positive assessment of our review.

1- It is somewhat ATL- and Japan-centered, it would benefit from a more global TCL and geographical approach

Response:

Thank you for your comment. In accordance with your suggestion, we have added the discussion to include cutaneous T-cell lymphomas (CTCL) and AITL/Tfh subtypes, and incorporated data from large international cohorts such as CIBMTR and SFGM-TC as follow:

Page 2, Line 52,

For example, in advanced-stage MF and SS, multiple prospective and retrospective studies have demonstrated the efficacy of allo-HCT, with 3-year overall survival rates reaching 40–60% [12-14].

Page 3, Line 54,

In AITL, an analysis of 249 patients with refractory disease or relapse after autologous transplantation reported 4-year PFS and OS rates of 49% and 56%, respectively [15].

Page 3, Line 65,

In the context of transplantation for TCLs, achieving optimal disease control prior to allo-HCT is one of the most critical factors of outcome [15,28]. Given the activity of these novel agents in TCLs, they are increasingly being utilized as salvage therapy for chemotherapy-refractory disease or as a bridge to allo-HCT. Furthermore, unlike conventional cytotoxic agents, some of these drugs have relatively limited myelosuppression, making them feasible for use as post-transplant maintenance therapy aimed at reducing the risk of relapse.

We have added the following references in conjunction with the revision.

  1. Weng, W.K.; Arai, S.; Rezvani, A.; Johnston, L.; Lowsky, R.; Miklos, D.; Shizuru, J.; Muffly, L.; Meyer, E.; Negrin, R.S.; et al. Nonmyeloablative allogeneic transplantation achieves clinical and molecular remission in cutaneous T-cell lymphoma. Blood Adv 2020, 4, 4474-4482, doi:10.1182/bloodadvances.2020001627.
  2. Domingo-Domenech, E.; Duarte, R.F.; Boumedil, A.; Onida, F.; Gabriel, I.; Finel, H.; Arcese, W.; Browne, P.; Beelen, D.; Kobbe, G.; et al. Allogeneic hematopoietic stem cell transplantation for advanced mycosis fungoides and Sézary syndrome. An updated experience of the Lymphoma Working Party of the European Society for Blood and Marrow Transplantation. Bone Marrow Transplant 2021, 56, 1391-1401, doi:10.1038/s41409-020-01197-3.
  3. Goyal, A.; O'Leary, D.; Foss, F. Allogeneic stem cell transplant for treatment of mycosis fungoides and Sezary syndrome: a systematic review and meta-analysis. Bone Marrow Transplant 2024, 59, 41-51, doi:10.1038/s41409-023-02122-0.

  1. Epperla, N.; Ahn, K.W.; Litovich, C.; Ahmed, S.; Battiwalla, M.; Cohen, J.B.; Dahi, P.; Farhadfar, N.; Farooq, U.; Freytes, C.O.; et al. Allogeneic hematopoietic cell transplantation provides effective salvage despite refractory disease or failed prior autologous transplant in angioimmunoblastic T-cell lymphoma: a CIBMTR analysis. J Hematol Oncol 2019, 12, 6, doi:10.1186/s13045-018-0696-z.

  1. Mamez, A.C.; Dupont, A.; Blaise, D.; Chevallier, P.; Forcade, E.; Ceballos, P.; Mohty, M.; Suarez, F.; Beguin, Y.; Peffault De Latour, R.; et al. Allogeneic stem cell transplantation for peripheral T cell lymphomas: a retrospective study in 285 patients from the Societe Francophone de Greffe de Moelle et de Therapie Cellulaire (SFGM-TC). J Hematol Oncol 2020, 13, 56, doi:10.1186/s13045-020-00892-4.

2- Including some reference to more standard agents would be helpful to clinicians (bexarotene, IFNa, methotrexate, ECP, CHOP, checkpoint inhibitors, etc), following Table 1 format

Response:

Thank you very much for this thoughtful comment. We also received similar feedback from another reviewer regarding immune checkpoint inhibitors and standard salvage therapy. Following the reviewers’ comments, we have added a new section on immune checkpoint inhibitors, as outlined below. In addition, we have updated Table 1 to include the impact of standard salvage chemotherapy and immune checkpoint inhibitors administered before and after allo-HCT on transplant outcomes.

Page 21, Line 382,

Immune checkpoint inhibitors (ICIs)

The activation of cytotoxic T cells requires not only signals from T cell receptors that recognize peptides presented on HLA molecules, but also co-stimulatory signals mediated by activating co-stimulatory molecules such as CD28. In contrast, some co-stimulatory molecules, such as CTLA-4 and PD-1, function as inhibitory molecules and suppress T cell activation. The pathways mediated by these inhibitory co-stimulatory molecules are known as immune checkpoints and play a critical role in tumor immune evasion [124]. Antibody drugs targeting these immune checkpoints, known as immune checkpoint inhibitors (ICIs), have been developed and have come into widespread use over the past decade for various treatment-resistant cancers [125].

Clinical trials are currently underway for various subtypes of TCLs [126]. Notably, promising treatment outcomes have been reported for extranodal NK/T-cell lymphoma, particularly with the use of pembrolizumab as salvage therapy in relapsed or refractory cases, with several studies demonstrating high response rates and durable remissions in a subset of patients [27,127]. On the other hand, there have been reports of rapid disease progression following nivolumab administration in indolent ATL, suggesting that caution may be needed when using ICIs in certain types of TCLs [128].

Impact of pre- and post-HCT use

Owing to their demonstrated efficacy as salvage therapy, ICIs are anticipated to serve as an effective bridging strategy prior to allo-HCT. However, based on reports focusing on Hodgkin lymphoma, the use of ICIs before allo-HCT is associated with an increased risk of GVHD [129-132]. Immune profiling suggested persistent immune alterations, including reduced PD-1+ T cells and lower Treg-to-effector T cell ratios [133]. Interestingly, several studies have reported that the increased risk of GVHD associated with the use of ICIs prior to allo-HCT may be mitigated by the use of PTCy as GVHD prophylaxis [134,135]. In allo-HCT for patients previously treated with ICIs, careful optimization of GVHD prophylaxis is of critical importance. On the other hand, the use of ICIs after allo-HCT has the potential to mitigate tumor immune evasion and T-cell exhaustion, thereby enhancing the GVL effect [136]. However, in such cases, careful monitoring is warranted not only for GVHD but also for immune-related adverse events associated with ICIs.

In addition, corresponding updates have been made to the abstract, introduction, and Table 1 to incorporate this information.

Page 1, Line 26 (Abstract),

… histone deacetylase inhibitors, enhancer of zeste homolog inhibitors, and immune checkpoint inhibitors—before or after allo-HCT…

Page 3, Line 64,

… enhancer of zeste homolog 1/2 (EZH1/2) inhibitors (e.g., valemetostat [26]), and immune checkpoint inhibitors (ICIs) (e.g., pembrolizumab [27]).

We have added the following references in conjunction with the revision.

  1. Kwong, Y.L.; Chan, T.S.Y.; Tan, D.; Kim, S.J.; Poon, L.M.; Mow, B.; Khong, P.L.; Loong, F.; Au-Yeung, R.; Iqbal, J.; et al. PD1 blockade with pembrolizumab is highly effective in relapsed or refractory NK/T-cell lymphoma failing l-asparaginase. Blood 2017, 129, 2437-2442, doi:10.1182/blood-2016-12-756841.

  1. Chen, L.; Flies, D.B. Molecular mechanisms of T cell co-stimulation and co-inhibition. Nature reviews. Immunology 2013, 13, 227-242, doi:10.1038/nri3405.

  1. Mc Neil, V.; Lee, S.W. Advancing Cancer Treatment: A Review of Immune Checkpoint Inhibitors and Combination Strategies. Cancers 2025, 17, 1408.

  1. Chen, X.; Wu, W.; Wei, W.; Zou, L. Immune Checkpoint Inhibitors in Peripheral T-Cell Lymphoma. Front Pharmacol 2022, 13, 869488, doi:10.3389/fphar.2022.869488.

  1. Li, X.; Cheng, Y.; Zhang, M.; Yan, J.; Li, L.; Fu, X.; Zhang, X.; Chang, Y.; Sun, Z.; Yu, H.; et al. Activity of pembrolizumab in relapsed/refractory NK/T-cell lymphoma. J Hematol Oncol 2018, 11, 15, doi:10.1186/s13045-018-0559-7.

  1. Ratner, L.; Waldmann, T.A.; Janakiram, M.; Brammer, J.E. Rapid Progression of Adult T-Cell Leukemia-Lymphoma after PD-1 Inhibitor Therapy. N Engl J Med 2018, 378, 1947-1948, doi:10.1056/NEJMc1803181.

  1. Casadei, B.; Broccoli, A.; Stefoni, V.; Pellegrini, C.; Marangon, M.; Morigi, A.; Nanni, L.; Lolli, G.; Carella, M.; Argnani, L.; et al. PD-1 blockade as bridge to allogeneic stem cell transplantation in relapsed/refractory Hodgkin lymphoma patients: a retrospective single center case series. Haematologica 2019, 104, e521-e522, doi:10.3324/haematol.2019.215962.

  1. Ijaz, A.; Khan, A.Y.; Malik, S.U.; Faridi, W.; Fraz, M.A.; Usman, M.; Tariq, M.J.; Durer, S.; Durer, C.; Russ, A.; et al. Significant Risk of Graft-versus-Host Disease with Exposure to Checkpoint Inhibitors before and after Allogeneic Transplantation. Biol Blood Marrow Transplant 2019, 25, 94-99, doi:10.1016/j.bbmt.2018.08.028.

  1. Soiffer, R.J. Checkpoint inhibition to prevent or treat relapse in allogeneic hematopoietic cell transplantation. Bone Marrow Transplant 2019, 54, 798-802, doi:10.1038/s41409-019-0617-y.

  1. Bobillo, S.; Nieto, J.C.; Barba, P. Use of checkpoint inhibitors in patients with lymphoid malignancies receiving allogeneic cell transplantation: a review. Bone Marrow Transplant 2021, 56, 1784-1793, doi:10.1038/s41409-021-01268-z.

  1. Merryman, R.W.; Kim, H.T.; Zinzani, P.L.; Carlo-Stella, C.; Ansell, S.M.; Perales, M.A.; Avigdor, A.; Halwani, A.S.; Houot, R.; Marchand, T.; et al. Safety and efficacy of allogeneic hematopoietic stem cell transplant after PD-1 blockade in relapsed/refractory lymphoma. Blood 2017, 129, 1380-1388, doi:10.1182/blood-2016-09-738385.

  1. Merryman, R.W.; Castagna, L.; Giordano, L.; Ho, V.T.; Corradini, P.; Guidetti, A.; Casadei, B.; Bond, D.A.; Jaglowski, S.; Spinner, M.A.; et al. Allogeneic transplantation after PD-1 blockade for classic Hodgkin lymphoma. Leukemia 2021, 35, 2672-2683, doi:10.1038/s41375-021-01193-6.

  1. Hu, Y.; Wang, Y.; Min, K.; Zhou, H.; Gao, X. The influence of immune checkpoint blockade on the outcomes of allogeneic hematopoietic stem cell transplantation. Frontiers in immunology 2024, 15, 1491330, doi:10.3389/fimmu.2024.1491330.

  1. Norde, W.J.; Maas, F.; Hobo, W.; Korman, A.; Quigley, M.; Kester, M.G.; Hebeda, K.; Falkenburg, J.H.; Schaap, N.; de Witte, T.M.; et al. PD-1/PD-L1 interactions contribute to functional T-cell impairment in patients who relapse with cancer after allogeneic stem cell transplantation. Cancer research 2011, 71, 5111-5122, doi:10.1158/0008-5472.Can-11-0108.

3- In the antibody drugs, some mention should be included of trials targeting other surface T-cell molecules (TRBC1, CD7, etc), such as PMID: 38538786/39528665, 38657244...

Response:

Thank you for your insightful comment. We agree that this is a highly important point, and in response, we have added the following statements regarding bispecific antibodies and CAR- T and NK as a future direction.

Page 23, Line 417,

Furthermore, novel T-cell and NK-cell immunotherapies—such as bispecific antibodies, chimeric antigen receptor (CAR)-engineered T cells or CAR-engineered NK cells targeting CD7 [137], CD30 [138-140], CD70 [141], and T-cell receptor beta constant region 1 (TRBC1) [142,143]— are currently under active de-velopment for TCLs.

We have added the following references in conjunction with the revision.

  1. Hu, Y.; Zhang, M.; Yang, T.; Mo, Z.; Wei, G.; Jing, R.; Zhao, H.; Chen, R.; Zu, C.; Gu, T.; et al. Sequential CD7 CAR T-Cell Therapy and Allogeneic HSCT without GVHD Prophylaxis. N Engl J Med 2024, 390, 1467-1480, doi:10.1056/NEJMoa2313812.

  1. Moskowitz, A.; Harstrick, A.; Emig, M.; Overesch, A.; Pinto, S.; Ravenstijn, P.; Schlüter, T.; Rubel, J.; Rebscher, H.; Graefe, T.; et al. AFM13 in Combination with Allogeneic Natural Killer Cells (AB-101) in Relapsed or Refractory Hodgkin Lymphoma and CD30 + Peripheral T-Cell Lymphoma: A Phase 2 Study (LuminICE). Blood 2023, 142, 4855-4855, doi:10.1182/blood-2023-174250.

  1. Grover, N.S.; Hucks, G.; Riches, M.L.; Ivanova, A.; Moore, D.T.; Shea, T.C.; Seegars, M.B.; Armistead, P.M.; Kasow, K.A.; Beaven, A.W.; et al. Anti-CD30 CAR T cells as consolidation after autologous haematopoietic stem-cell transplantation in patients with high-risk CD30+ lymphoma: a phase 1 study. The Lancet Haematology 2024, 11, e358-e367, doi:https://doi.org/10.1016/S2352-3026(24)00064-4.

  1. Nieto, Y.; Banerjee, P.; Kaur, I.; Basar, R.; Li, Y.; Daher, M.; Rafei, H.; Kerbauy, L.N.; Kaplan, M.; Marin, D.; et al. Allogeneic NK cells with a bispecific innate cell engager in refractory relapsed lymphoma: a phase 1 trial. Nature Medicine 2025, 31, 1987-1993, doi:10.1038/s41591-025-03640-8.

  1. Iyer, S.P.; Sica, R.A.; Ho, P.J.; Prica, A.; Zain, J.; Foss, F.M.; Hu, B.; Beitinjaneh, A.; Weng, W.-K.; Kim, Y.H.; et al. Safety and activity of CTX130, a CD70-targeted allogeneic CRISPR-Cas9-engineered CAR T-cell therapy, in patients with relapsed or refractory T-cell malignancies (COBALT-LYM): a single-arm, open-label, phase 1, dose-escalation study. The Lancet Oncology 2025, 26, 110-122, doi:10.1016/S1470-2045(24)00508-4.

  1. Nichakawade, T.D.; Ge, J.; Mog, B.J.; Lee, B.S.; Pearlman, A.H.; Hwang, M.S.; DiNapoli, S.R.; Wyhs, N.; Marcou, N.; Glavaris, S.; et al. TRBC1-targeting antibody-drug conjugates for the treatment of T cell cancers. Nature 2024, 628, 416-423, doi:10.1038/s41586-024-07233-2.

  1. Cwynarski, K.; Iacoboni, G.; Tholouli, E.; Menne, T.; Irvine, D.A.; Balasubramaniam, N.; Wood, L.; Shang, J.; Xue, E.; Zhang, Y.; et al. TRBC1-CAR T cell therapy in peripheral T cell lymphoma: a phase 1/2 trial. Nat Med 2025, 31, 137-143, doi:10.1038/s41591-024-03326-7.

Reviewer 3 Report

Comments and Suggestions for Authors

This review provides a clear description of the recent evidence on how novel agents for T cell lymphoma impact allogeneic transplantation.

The impact of each novel agent on GVHD is described, and this is useful information for clinicians. Thus, it is scientifically sound and contains sufficient interest and originality to merit publication.

If possible, it would be better the text could be more compact.

Author Response

This review provides a clear description of the recent evidence on how novel agents for T cell lymphoma impact allogeneic transplantation.

The impact of each novel agent on GVHD is described, and this is useful information for clinicians. Thus, it is scientifically sound and contains sufficient interest and originality to merit publication.

Response:

We are sincerely grateful for your very positive and encouraging feedback on our review.

If possible, it would be better the text could be more compact.

Response:

Thank you for your comment. In accordance with your suggestion, we have substantially removed data unrelated to TCLs regarding the impact of pre-transplant LEN administration and have revised the section to present a more concise summary, as follows:

Page 13, Line 240,

LEN administration early after allo-HCT has been associated with a high risk of severe GVHD [81,82], due to rapid proliferation of donor-derived T cells [83]. Delaying initiation until ≥3–6 months post-transplant and/or dose reduction has been reported to mitigate this risk [84-87]. Moreover, combining LEN with azacitidine (AZA), which can expand Tregs, may enhance anti-tumor activity without increasing GVHD incidence [88,89]. Nonetheless, cytopenia remain a major limitation, requiring careful dose and schedule adjustment.

Round 2

Reviewer 1 Report

Comments and Suggestions for Authors

Thank you for submitting the revised manuscript.

Most of the comments have been appropriately addressed.

2 minor issues 

  1. Page 10 line 168: "However, reports on the use of MOG after allo-HCT are limited to 
    small retrospective series, and further investigations from both basic and clinical perspectives are required." This statement regarding the risks of GVHD, applies to the use of MOG in both the pre- and post-transplant settings.  
  2.  Table 1 is titled " Summary of the impact of Novel agents on..." but now also includes chemotherapy. It would be more appropriate to title it " Summary of the impact of salvage therapy on..."

Author Response

Reviewer: 1 - Round 2 –

Thank you for submitting the revised manuscript.

Most of the comments have been appropriately addressed.

Response:

We are sincerely grateful for your detailed review of our revised manuscript.

2 minor issues

  1. Page 10 line 168: "However, reports on the use of MOG after allo-HCT are limited to

small retrospective series, and further investigations from both basic and clinical perspectives are required." This statement regarding the risks of GVHD, applies to the use of MOG in both the pre- and post-transplant settings. 

Response:

Thank you for pointing it out. Following the reviewer’s comment, we revised the content as follows:

Page 9, Line 168,

However, reports on the use of MOG in pre- and post-transplant settings are limited to small retrospective series, and further investigations from both basic and clinical perspectives are required.

  1. Table 1 is titled " Summary of the impact of Novel agents on..." but now also includes chemotherapy. It would be more appropriate to title it " Summary of the impact of salvage therapy.."

Response:

Thank you for your suggestion. Following the reviewer’s suggestion, we revised the title of Table 1 as follows:

Page 22, Line 425 (Table 1),

Table 1. Summary of the Impact of Salvage Therapies on T-Cell Lymphomas Pre- and Post-Allogeneic Hem-atopoietic Cell Transplantation.